# Association of food environment with diet quality and Body Mass Index (BMI) of school-going adolescents in Nepal

**Pragya Sharma**[1]*, **Neha Limaye**[2], **Rajeeb Kumar Sah**[3], **Archana Shrestha**[1,4]

1 Department of Public Health, Kathmandu University School of Medical Sciences, Dhulikhel, Nepal,
2 Department of Medicine, Mount Sinai Hospital, New York, United States of America, 3 School of Human and Health Sciences, University of Huddersfield, Huddersfield, United Kingdom, 4 Institute for Implementation Science and Health, Kathmandu, Nepal

* psharmaa973@gmail.com

## Abstract

### Background

Adolescents are being more vulnerable to non-communicable diseases (NCDs). A healthy food environment is crucial in maintaining a healthy diet and achieving better health outcomes. This study aimed to determine how certain features of home food environment affect diet quality and Body Mass Index (BMI) of school-going adolescents.

### Methods

We conducted a cross-sectional analytical study among 678 school-going adolescents aged 15–19 years in Budanilkantha municipality of Kathmandu, Nepal. We assessed home availability of food items in the past day, walking time needed to reach nearest shops from home, parental modeling, and parenting style. Furthermore, we assessed diet quality using a Diet Quality Questionnaire as Global Dietary Recommendations (GDR) Score and measured height and weight of participants to calculate BMI. We used multiple regression models to analyze data, all statistical analyses were performed using STATA-14.

### Results

Of 678 participants, 51.92% were males, and mean age was 15.56 years. Those who had to walk > 20 minutes to reach nearest vegetable shop had an average 1.44 point lower GDR Score (95% CI: −2.08, −0.19) than those with vegetable shops at their home. Those who had processed meat at home in the past day had 1.61 points lower GDR Score (95% CI: −1.95, −1.28), those with fruits and vegetables had 0.74 points lower GDR Score (95% CI: 0.48, 1.00) as compared to those who didn't have. Participants who had starchy staple available had a BMI score lower by 5.59 kg/m² on average (95% CI: − 10.78, − 0.40), and when two participants whose parental modeling scores differed by a unit were compared, the one with a higher score had on an average 0.19 kg/m² greater BMI (95% CI: 0.01, 0.37).

**Data availability statement:** All relevant data are within the paper and its Supporting Information files.

**Funding:** The author(s) received no specific funding for this work.

## Conclusion

This study highlights impact of home food environment on diet quality and BMI among adolescents in urban Nepal. Availability of healthy foods positively affects diet quality, while unhealthy items negatively influences it. Future research should explore wider food environmental factors and intervention strategies to improve adolescents' diet quality.

## Introduction

Non-communicable diseases (NCDs) accounted for 27% of deaths and 53% of Disability Adjusted Life years (DALYs) among adolescents globally [1]. Approximately 70% of adult-onset premature deaths are caused by health-related behaviors that start during childhood and adolescence [2]. Diet and Non-Communicable Diseases (NCDs) have an intricate relationship, with dietary choices crucial in determining chronic health conditions [3].

The global impact of poor diet is significant, w ith one in every five fatalities worldwide and approximately 11 million deaths attributed to an unhealthy diet [4]. More than half of these diet-related deaths are associated with high sodium consumption and insufficient consumption of healthy grains and fruits. Junk foods, defined by their lack of nutrition, vitamins, and minerals, coupled with high levels of processing, excessive salts, sugars, and fats, pose a significant health risk [5], with 70% of the global population consuming them [6].

Inadequate nutrient intake, including micronutrient deficiencies, contributes to the development of NCDs [7]. At the same time, excessive caloric consumption, particularly from energy-dense, nutrient-poor foods, is a significant factor in obesity-related NCDs such as cardiovascular diseases and diabetes [8]. Specific dietary components, such as high sodium, saturated fats, and added sugars, are associated with elevated risks of hypertension, heart disease, and metabolic disorders whereas dietary patterns like the Mediterranean and Dietary Approaches to Stop Hypertension (DASH) diets, rich in fruits, vegetables, and whole grains, demonstrate preventive effects on NCDs [9].

The food environment influences food acquisition and consumption of the individuals. It includes external factors such as food availability and price as well as individual determinants like accessibility and affordability [10]. Healthy food environments facilitate an individual's ability to make healthy food choices, enhancing overall dietary quality and improving health [11]. However, unhealthy options can influence individuals to purchase and consume unhealthy foods [12]. Various socio-cognitive and socio-ecological theories have suggested that distinct elements of the home food environment, such as parental practices and food availability at home, shape the dietary practices of adolescents [13]. The home food environment is increasingly recognized as a critical determinant influencing food preferences and eating behaviors, contributing to obesogenic situations [14]. Several studies have demonstrated the relationship between an unhealthy food environment and obesity, chronic disease, and other health-related factors [15–17]. Studies on the food environment have been conducted predominantly in the setting of High-Income Countries (HICs) [12]. However, a significant research gap exists in the context of Low- and Middle-Income Countries (LMICs), including Nepal, particularly among adolescents [10].

The Nepal Demographic and Health Survey 2022 reports that among adolescent girls aged 15–19 years, 27% are of short stature (indicating stunting), 26% are thin (indicating undernutrition), 5% are overweight, and less than 1% are obese, while among adolescent boys of the same age group, 7% are of short stature (indicating stunting), 41% are thin (indicating undernutrition), 4% are overweight, and 3% are obese [25]. Adolescent Nutrition Survey in

Nepal, 2014 indicated that 42.2% of adolescents aged 15–19 years were underweight, 55.9% had a normal weight and only 1.8% were overweight [18].

The double burden of malnutrition is rising among Nepalese adolescents [25]. GDR score is an easy to calculate indicator for the assessment of their diet quality. The score indicates adherence to global dietary recommendations, which include dietary factors protective against non-communicable diseases. The higher the GDR score, the more recommendations are likely to be met [22]. Understanding diet quality in this population is essential, as it informs targeted interventions aimed at improving nutrition and addressing the growing concerns of both underweight and overweight adolescents in Nepal.

This study assessed the association of home food environment, including home food availability, accessibility to shops, parental modeling, and parenting style, with diet quality and Body Mass Index (BMI) among school-going adolescents in Budanilkantha municipality in Kathmandu, Nepal. The findings of the study will help policymakers and concerned authorities take action to improve dietary intake and prevent diet-related diseases.

## Methods

### Study design and site

We conducted a cross-sectional study among adolescents attending schools in Budanilkantha Municipality in central Nepal. The municipality has a total population of 153,203 [19] and the total number of adolescents aged 10–19 years is 29,651 according to census 2021 [20]. The municipality has 19 public, 113 private, and 4 religious schools [19]. Out of these, we identified 84 schools that were operating secondary level classes at the time of data collection.

### Participants

Study participants included individuals aged 15–19 years who were enrolled in secondary-level classes in schools located in Budanilkantha municipality. Students living in hostels, and those unable to communicate were excluded from the study. We conducted two-stage sampling to recruit the participants. In the first stage, we randomly selected 29 schools from a list of 84 schools. In the second stage, we randomly selected 24 eligible students from each school. We enrolled 689 participants from 17th June to 27th September, 2022, 11 of them were non-responsive making our final sample size 678.

### Sample size

To determine sample size, we followed three consecutive steps. Initially, we calculated a sample size of 447 using the formula $n = (Z_{\alpha/2}+Z_{\beta})^2 \, 2\sigma^2/ d^2$. Here, $Z_{\alpha/2}$ (1.96) is the critical value of the Normal distribution at 95%, with α set at 0.05 for the significance level. $Z_{\beta}$ (0.84) is the critical value at β (0.2) for an 80% power, σ2 (2.56) is the population variance, and d (0.3) is the mean difference in global dietary recommendation score between those with and without access to processed meat at home, obtained from preliminary studies. Considering clustering within schools, we adjusted the sample size by multiplying it by the design effect (1.285), resulting in 574. The design effect was calculated as $1 + \delta (n - 1)$, where δ (ICC) was 0.015, and n =20 was the cluster size [21]. Finally, we factored in a 20% non-response rate.

### Ethical clearance

We obtained ethical clearance from Institutional Review Board (IRB) at Kathmandu University School of Medical Sciences (approval number 38/22). Prior to engaging with participants, we secured permission letter from Budanilkantha municipality office, along with permission

from administrations of all participating schools. Following selection of eligible participants, we conducted 15-minute orientation session to acquaint them with the study. Participants aged 18 years and above were provided the consent form, whereas for those under 18, we provided a concise description of study's objectives and what was expected of participants, along with a written assent form to be signed by their parents.

## Outcome assessment

**Diet quality.** We assessed our primary outcome, the diet quality using the Global Dietary Recommendations (GDR) Score, derived from the Diet Quality Questionnaire (DQQ) [22]. The DQQ is a standardized tool for assessment of diet quality based on consumption of different food items belonging to different food groups (Table 1) in the past day and has been adapted for Nepal. The food intake data is used to estimate the GDR score, which ranges from 0 to 18. This score serves as an indicator of adherence to global dietary guidelines aimed at mitigating non-communicable diseases. A higher GDR score indicates a greater likelihood of meeting healthy dietary recommendations [22]. GDR score is calculated as (GDR-Healthy) - (GDR - Limit) + 9. The GDR-Healthy score, a component of GDR score ranges from 0 to 9, is calculated by adding the scores ("0" and "1") obtained in questions indicating consumption of health-promoting foods in the past day and correlates positively with meeting global dietary recommendations. The GDR-Limit score, another component ranging from 0 to 9, is calculated by adding the scores ("0" and "1") obtained in questions indicating consumption of

**Table 1. Socio-demographic characteristics of the participants (n = 678) - Budanilkantha, Nepal, 2022.**

| Variables | Frequency (Percentage) |
|---|---|
| Grade | |
| Nine | 262 (38.6) |
| Ten | 416 (61.4) |
| Gender | |
| Female | 326 (48.1) |
| Male | 352 (51.9) |
| Age in years (Mean ± SD) | 15.6 ± 0.7 |
| Ethnicity | |
| Adibasi/Janajati | 342 (50.4) |
| Brahmin/Chhetri | 255 (37.6) |
| Others | 81 (11.6) |
| Religion | |
| Hinduism | 481 (70.9) |
| Buddhism | 136 (20.1) |
| Others | 61 (8.6) |
| Media Sources | |
| Newspaper | 103 (15.2) |
| Television | 378 (55.8) |
| Radio | 43 (6.3) |
| Internet | 657 (96.9) |
| Family type | |
| Joint | 172 (25.4) |
| Nuclear | 506 (74.6) |
| Number of family members (n, Mean ± SD) | (669, 5.1 ± 2.2) |

components of the diet to limit or avoid in the past day and correlates negatively with meeting global dietary recommendations [22].

**Body mass index (BMI).** We measured participants' weight after they removed their shoes and outerwear using the OMRON-400. We measured their height using a standard measuring tape with participants standing against a wall for measurement [23]. We calculated BMI as weight in kilograms divided by height in meters squared. BMI was the secondary outcome of our study.

### Exposure assessment

The following four dimensions of home food environment, a) Home food availability, b) Accessibility to different types of shops, c) Parental Modeling and d) Parenting Style were assessed using a self-administered questionnaire:

**a. Home food availability.** We measured the availability of nine groups of food, consisting of 29 specific food items (as shown in S1 Table) at home, based on the DQQ Indicator Guide [22]. The participants checked "yes" if at least one food item within the group was present at home on the previous day of data collection.

**S1 Table: Food groups and the corresponding items** [22].(WORD)**b. Accessibility to different types of shops.** Participants reported walking time (in minutes) to reach the nearest grocery store, fruit shop, vegetable shop, and street food shop from the participant's homes. We estimated walking times for these four types of vendors because of variations in the foods they offer. Typically, grocery stores offer stock items such as rice, pulses, flour, and packaged foods like instant noodles, biscuits, chocolates, and chips. Fruit shops offer seasonal fruits, but apples and bananas are consistently available. Vegetable shops primarily offer seasonal vegetables, yet potatoes, tomatoes, and onions are consistently stocked. Street food shops usually sell *chatpate* (a spicy Nepali snack and can be unhealthy due to its high salt, oil, spices content and potential contamination from unhygienic preparation methods), *panipuri* (a popular Nepali street snack, often unhealthy due to its deep-fried *puris*, potentially unhygienic preparation conditions and unhealthy spices), *samosa* (a deep-fried Nepali snack filled with spiced potatoes, often unhealthy due to its high fat content, refined flour and potential oil reuse) and *pakauda* (deep-fried snacks often unhealthy due to high oil content and potential reuse of oil).

**c. Accessibility.** To measure accessibility to shops, based on the participants' reported walking time to different types of shops (grocery, fruits shop, vegetables shop and street food shops) from their home, we used the concept of "10 minute city" [24] to categorize the data into the following four categories: (i) 0 minutes (for those who had a particular type of shop at their own home), (ii) less than 10 minutes, (iii) between 10 and 20 minutes, and (iv) more than 20 minutes.

**d. Parental modeling and parental style.** To assess parental modeling and parenting style, we adapted questions involving adolescent-parent pairs in the US who participated in an e-health lifestyle modification program [13]. We obtained permission to use the tool from the original author, translated it into Nepali, and back-translated it by two independent native Nepali speakers proficient in English. The internal consistency of the tool, as indicated by Cronbach's alpha in our pilot study, was 0.73. The tool assessed parental modeling through four statements on the food items the parents eat when around their children rated on a 4-point Likert scale (Never, Sometimes, Often, and Always). The tool evaluated parenting style through sixteen statements on their child's problems, awareness of their child's whereabouts after school, and unconditional acceptance of their child. Responses were also rated on a 4-point scale, with reverse coding applied to five statements reflecting negative parenting behavior. Higher score denoted more favorable parenting style and parenting modeling.

## Other co-variates

We assessed socio-demographic characteristics of participants (age, gender, ethnicity, religion, family type, and size) and participants' parents (Marital status, Employment Status, Occupation, Education and Family Income status) using Nepal Demographic Health Survey questionnaires [25]. All these socio-demographic variables have been included for adjustment in model 2 and 3 of data analyses. We assessed physical activity using International Physical Activity Questionnaire [26] and estimated weekly MET minutes. We incorporated questions from the World Health Organization STEPs survey to assess smoking status and classified participants as 'never,' 'former,' and 'current' smokers.

A conceptual framework was developed to guide the study, illustrating relationship between home food environment, diet quality, and BMI among adolescents. The framework allowed identification of the variables and their interaction among each other. The conceptual model (Fig 1) illustrates hypothesized relationships among home food environment, diet quality, BMI and other covariates that include socio-demographic characteristics and lifestyle factors. It also guided the selection of variables for data collection and informed the data analysis by providing a structured approach to examine the interactions among the assessed variables.

## Statistical analysis

We generated descriptive statistics to describe participants' socio-demographic characteristics, Global Dietary Recommendations (GDR) Score, and BMI using mean and standard deviation for continuous variables and frequency and percentage for categorical variables. We utilized multivariate linear regression to assess the association of the two outcomes: GDR score and BMI ($kg/m^2$), with four exposure variables: food availability of nine food categories at home in the past day, accessibility in terms of walking time to food vendors, parenting style, and parenting mode in 4-point Likert scale. In all models, we adjusted for socio-demographic variables and lifestyle (smoker vs. non-smokers and MET minutes per week). The adjustment variables were selected apriori based on literature review. We tested and confirmed that all regression models met four assumptions of linear regression, linear relationship, independence (by design), homoscedasticity, and normality. All statistical analyses were performed using STATA -14.

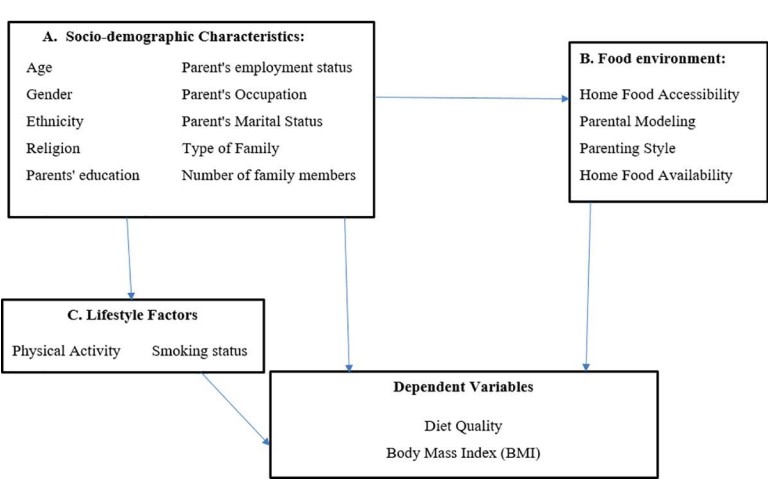

**Fig 1. Conceptual Framework of the study.**

## Results

Table 1 presents socio-demographic characteristics of participants. Majority of the participants were in tenth grade (61%), with a balanced distribution between genders. The mean age was 15.6 (SD=0.7) years. The participants were predominantly of Adibasi/Janajati ethnicity (50%). Internet was the most popular media source (97%). Majority came from nuclear family structure (75%), and average family size was 5.1. Table 2 presents characteristics of the participant's parents. Majority of parents were living together (78%). Most fathers (81%) were employed full-time, whereas nearly half of the mothers (48%) were unemployed. Fathers' occupations varied, with 42% engaged in daily labor and most of the others being homemakers (49%). The majority of participants' fathers had completed higher secondary education (34.8%) or lower secondary education (31.4%). For mothers, the most common educational levels were literate (39.2%) and higher secondary education (26.4%).

The mean Global Dietary Recommendations (GDR) Score was 8.58, and the minimum and maximum scores were 2 and 13, respectively. The majority of our study participants were either underweight (41%) or had a normal weight (47%) as shown in Fig 2.

Table 3 presents the bivariate and multivariate models to assess the association of GDR score with food accessibility, food availability, parenting style, and parenting modeling. Accessibility to vegetable shops, street food vendors, and grocery shops was associated with the GDR score. Specifically, in comparison to individuals with vegetable shops conveniently located at their homes, those who can access the nearest vegetable shop in less than 10 minutes had a GDR Score around 0.88 points lower (95% CI: −1.62, −0.13, p-value = 0.02); and those requiring over 20 minutes to reach the nearest vegetable shop had a GDR Score 1.44 points lower (95% CI: −2.08, −0.19, p-value = 0.02). Participants who must walk over 20 minutes to access the closest street food vendor had a 0.69 points lower GDR score than individuals who had a street food vendor at their residence (95% CI: −1.18, −0.20, p-value < 0.01). Individuals who could walk to the nearest grocery shop within 10–20 minutes had 0.94 points higher GDR Score than those with a grocery shop located at their home (95% CI: 0.25, 1.63, p-value < 0.01). Accessibility to fruit shops was not associated with the GDR score.

The presence of salty or fried snacks and processed meat at home the previous day was associated with a lower GDR score, while having fruits and vegetables at home was associated with a higher GDR score. Specifically, individuals reporting the presence of salty or fried snacks at home the previous day had an average GDR score of 0.44 points lower (95% CI: −0.78, −0.08, p-value = 0.02), those with processed meat had an average GDR score of 1.61 points lower (95% CI: −1.95, −1.28, p-value < 0.001) and those with fruits and vegetables at home had an average GDR score about 0.74 points higher (95% CI: 0.48, 1.00, p-value < 0.001) compared to those without these food items at home. Availability of other foods such as starchy staples, legumes, nuts, and seeds, animal source foods, fiber-containing foods, sugar-sweetened food items, and saturated fat-containing foods was not significantly associated with the GDR score. There was no significant association between the GDR score and parental modeling or parenting style in the bivariate or multivariate models.

Table 4 presents the bivariate and multivariate models to assess the association of body mass index with food accessibility, availability, parenting style, and parental modeling. Accessibility to vegetable shops was associated with BMI. Participants who required a 10–20 minute walk to access the nearest vegetable shop, on average, had a BMI of 2.44 kg/m2 higher (95% CI: 0.29, 4.59, p-value = 0.03), whereas those needing over 20 minutes had a BMI of 3.13 kg/m2 higher (95% CI: 0.55, 5.71, p-value = 0.02) compared to those with a vegetable shop at

**Table 2. Characteristics of the Participants' Parents (n=678) Budanilkantha, Nepal, 2022.**

| Variables | Frequency (Percentage) |
|---|---|
| Marital status of parents | |
| Parents living together | 531 (78.3) |
| Father lives away from home | 96 (14.2) |
| Others | 51 (7.5) |
| Father's Employment Status | |
| Unemployment | 53 (7.8) |
| Part-time Job | 78 (11.5) |
| Full-time Job | 547 (80.6) |
| Mother's Employment Status | |
| Unemployment | 323 (47.6) |
| Part-time Job | 69 (10.2) |
| Full-time Job | 286 (42.2) |
| Father's Occupation | |
| Daily Wages | 274 (40.4) |
| Private Job | 188 (27.7) |
| Foreign Employment | 73 (10.8) |
| Business | 73 (10.8) |
| Others | 70 (10.3) |
| Mother's Occupation | |
| Homemaker | 330 (48.7) |
| Private Job | 125 (18.4) |
| Business | 108 (15.9) |
| Others | 115 (16.9) |
| Father's education | |
| Illiterate | 8 (1.2) |
| Literate | 154 (22.7) |
| Lower Secondary level | 213 (31.4) |
| Higher Secondary Level | 236 (34.8) |
| Bachelor's and above | 67 (9.9) |
| Mother's education | |
| Illiterate | 26 (3.8) |
| Literate | 266 (39.2) |
| Lower Secondary level | 168 (24.8) |
| Higher Secondary Level | 179 (26.4) |
| Bachelor's and above | 39 (5.8) |
| Family Income | |
| Both of the parents working full time | 245 (36.1) |
| Only one parent working full-time | 78 (11.5) |
| Both the parents working part-time | 15 (2.2) |
| Single income family | 313 (46.2) |
| Not working at present time | 27 (3.9) |

home. Compared to individuals with starchy staples at home, those who didn't have starchy staples available the previous day had a BMI of 5.58 kg/m2 lower (95% CI: −10.76, −0.38, p-value = 0.04). When comparing BMI, participants whose parental modeling score differed by one unit had on an average 0.18 kg/m$^2$ higher BMI 95% CI: 0.008, 0.36, p-value = 0.04).

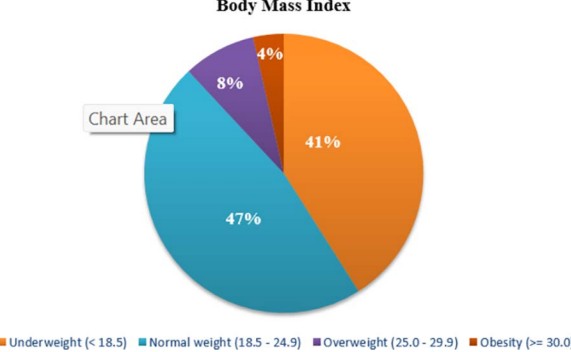

**Fig 2. Body Mass Index (BMI) Chart of the study participants.**

## Discussion

In an urban area in Nepal, the diet quality of adolescents is likely to be influenced by accessibility to different types of shops. Our findings indicate that those who have to walk longer distances to get to vegetable shops and street food vendors tend to have lower-quality diets. Additionally, having salty or fried snacks and processed meat available at home is associated with poorer diet quality among adolescents. Conversely, when fruits and vegetables are available at home, teenagers will likely have a better diet. Parenting modeling and parenting style are not related to adolescent's diet quality. Being close to vegetable shops and having starchy staples available at home is linked to lower body mass index (BMI). Positive parental modeling, meaning when parents demonstrate healthy eating behaviors, is associated with teenagers having higher BMI.

Adolescents had lower diet quality scores when they lived at a walking distance of less than 10 minutes or greater than 20 minutes to a vegetable shop; this association did not exist for accessibility of fruit shops. Studies [27,28] have documented that increased accessibility was positively associated with fruit and vegetable intake in children, but this was not seen in our findings. A number of contextual factors may explain this discrepancy. One possible explanation for lower diet quality associated with vegetable shops farther away is that these places may have fewer overall healthy food alternatives or be located in communities with limited access to fresh produce markets. Furthermore, greater distances to vegetable shops could exacerbate the challenges associated with transportation, time constraints, and economic barriers to purchasing fresh produce. This could lead to a general lack of access to a diverse range of fresh and healthy foods, not only vegetables, resulting in a lower quality diet.

Adolescents also had lower diet quality scores when they lived within 10 minutes or > 20 minutes from street food vendors. However, it's important to note that the study only measured the accessibility to street food shops from the adolescents' homes. It did not consider the distance from their schools, which could also play a significant role in their eating habits, as adolescents might spend a significant portion of their day at school [29] could purchase food from vendors near their school instead of near their home as the accessibility of street food in neighborhoods close to schools may have a big impact on their dietary choices [32].

In our study, adolescents who had home availability of salty or fried snacks and processed meat in the past day had lower diet quality. In comparison, those with the availability of fruits and vegetables had higher diet quality. Previous studies [30,31] have found similar associations between availability and adolescent consumption of various food items. Availability of fruits and vegetables at home is associated with a higher likelihood of their consumption,

**Table 3. Associations of Home Food Environment with Diet Quality (GDR Score) - Budanilkantha, Nepal, 2022.**

| Features of Home Food Environment | | *Model 1 | | | ** Model 2 | | | *** Model 3 | | |
|---|---|---|---|---|---|---|---|---|---|---|
| Accessibility | n (%) | Coefficient | 95% CI | p-value | Coefficient | 95% CI | p-value | Coefficient | 95% CI | p-value |
| Grocery Shop | | | | | | | | | | |
| 0 minute (Ref) | 67 (9.88) | | | | | | | | | |
| < 10 minutes | 564 (83.19) | 0.23 | (−0.21, 0.67) | 0.29 | 0.21 | (−0.23, 0.65) | 0.34 | 0.33 | (−0.10, 0.77) | 0.13 |
| 10–20 minutes | 39 (5.75) | 1.04 | (0.35, 1.73) | 0.003 | 1.11 | (0.42, 1.79) | 0.002 | 0.95 | (0.26, 1.65) | 0.007 |
| > 20 minutes | 8 (1.18) | −1.47 | (−2.75, 0.19) | 0.03 | −1.2 | (−2.56, −0.008) | 0.049 | −0.63 | (−1.86, 0.62) | 0.33 |
| Fruits Shop | | | | | | | | | | |
| 0 minute (Ref) | 13 (1.92) | | | | | | | | | |
| < 10 minutes | 430 (63.42) | −0.03 | (−1.01, 0.94) | 0.95 | −0.02 | (−1.00, 0.97) | 0.98 | 0.29 | (−0.73, 1.32) | 0.57 |
| 10–20 minutes | 184 (27.14) | −0.28 | (−1.28, 0.71) | 0.57 | −0.28 | (−1.29, 0.72) | 0.58 | −0.10 | (−1.14, 0.94) | 0.85 |
| > 20 minutes | 51 (7.52) | −0.18 | (−1.26, 0.89) | 0.74 | −0.28 | (−1.36, 0.81) | 0.61 | 0.06 | (−1.06, 1.18) | 0.92 |
| Vegetables Shop | | | | | | | | | | |
| 0 minute (Ref) | 24 (3.54) | | | | | | | | | |
| < 10 minutes | 500 (73.75) | −0.45 | (−1.17, 0.27) | 0.22 | −0.47 | (−1.19, 0.25) | 0.19 | −0.88 | (−1.62, −0.13) | 0.02 |
| 10–20 minutes | 124 (18.29) | −0.29 | (−1.06, 0.48) | 0.46 | −0.36 | (−1.14, 0.41) | 0.35 | −0.69 | (−1.48, −0.09) | 0.08 |
| > 20 minutes | 30 (4.42) | −0.77 | (−1.72, 0.18) | 0.11 | −0.96 | (−1.91, −0.006) | 0.049 | −1.14 | (−2.08, −0.19) | 0.01 |
| Street food Vendors | | | | | | | | | | |
| 0 minute (Ref) | 285 (42.04) | | | | | | | | | |
| < 10 minutes | 215 (31.71) | −0.32 | (−0.63, 0.007) | 0.04 | −0.30 | (−0.61, −0.01) | 0.06 | −0.29 | (−0.59, −0.001) | 0.05 |
| 10–20 minutes | 127 (18.73) | −0.32 | (−0.69, 0.04) | 0.08 | −0.39 | (−0.76, −0.03) | 0.04 | −0.31 | (−0.65, 0.03) | 0.08 |
| > 20 minutes | 51 (7.52) | −0.91 | (−1.43, 0.39) | 0.001 | −0.90 | (−1.42, −0.38) | 0.001 | −0.69 | (−1.18, −0.21) | 0.005 |
| Home Food Availability Starchy Staple | | | | | | | | | | |
| No (Ref) | 15 (2.21) | | | | | | | | | |
| Yes | 663 (97.79) | −0.43 | (−1.33, 0.48) | 0.35 | 0.36 | (−0.59, 1.31) | 0.46 | 0.03 | (−1.76, 1.79) | 0.97 |
| Legumes, Nuts & Seeds | | | | | | | | | | |
| No (Ref) | 32 (4.72) | | | | | | | | | |
| Yes | 646 (95.28) | 0.31 | (−0.31, 0.94) | 0.33 | 0.67 | (0.04, 1.30) | 0.04 | 0.58 | (−0.47, 1.64) | 0.28 |
| Animal-source foods | | | | | | | | | | |
| No (Ref) | 37 (5.46) | | | | | | | | | |
| Yes | 641 (94.54) | −0.18 | (−0.77, 0.40) | 0.53 | −0.009 | (−0.60, 0.58) | 0.98 | −0.03 | (−0.77, 0.69) | 0.93 |
| Salty or Fried Snacks | | | | | | | | | | |
| No (Ref) | 116 (17.11) | | | | | | | | | |
| Yes | 562 (82.89) | −0.52 | (−0.88, −0.17) | 0.003 | −0.39 | (−0.75, −0.04) | 0.03 | −0.44 | (−0.79, −0.08) | 0.02 |
| Processed meat | | | | | | | | | | |
| No (Ref) | 567 (83.63) | | | | | | | | | |
| Yes | 111 (16.37) | −1.59 | (−1.94, −1.26) | < 0.001 | −1.62 | (−1.97, −1.29) | < 0.001 | −1.61 | (−1.95, −1.28) | < 0.001 |
| Fruits & Vegetables | | | | | | | | | | |
| No (Ref) | 440 (64.90) | | | | | | | | | |
| Yes | 238 (35.10) | 0.76 | (0.49, 1.03) | < 0.001 | 0.68 | (0.41, 0.96) | < 0.001 | 0.74 | (0.48, 1.00) | < 0.001 |
| Fiber containing foods | | | | | | | | | | |
| No (Ref) | 23 (3.39) | | | | | | | | | |
| Yes | 655 (96.61) | 0.15 | (−0.58, 0.89) | 0.69 | 0.70 | (−0.05, 1.45) | 0.07 | 0.98 | (−0.54, 2.51) | 0.21 |
| Sugar-sweetened food items | | | | | | | | | | |
| No (Ref) | 30 (4.42) | | | | | | | | | |
| Yes | 648 (95.58) | −0.23 | (−0.87, 0.42) | 0.48 | −0.003 | (−0.65, 0.65) | 0.99 | −0.19 | (−1.03, 0.62) | 0.65 |

*(Continued)*

**Table 3.** (Continued)

| Features of Home Food Environment | | *Model 1 | | | ** Model 2 | | | *** Model 3 | | |
|---|---|---|---|---|---|---|---|---|---|---|
| Accessibility | n (%) | Coefficient | 95% CI | p-value | Coefficient | 95% CI | p-value | Coefficient | 95% CI | p-value |
| Saturated fat-containing foods | | | | | | | | | | |
| No (Ref) | 43 (6.34) | | | | | | | | | |
| Yes | 635 (93.66) | −0.57 | (−1.12, −0.03) | 0.04 | −0.45 | (−1.00, 0.10) | 0.11 | −0.52 | (−1.18, 0.15) | 0.13 |
| Parental Modeling | 678 | 0.03 | (−0.03, 0.09) | 0.35 | −0.004 | (0.07, 0.06) | 0.91 | −0.0003 | (−0.06, 0.06) | 0.99 |
| Parenting Style | 678 | 0.007 | (−0.02, 0.03) | 0.59 | 0.003 | (−0.02, 0.03) | 0.80 | −0.006 | (−0.03, 0.02) | 0.64 |

Outcome variable: Global Dietary Recommendations (GDR) score indicating the diet quality

*Model 1: Bivariate Analysis of Features of Home Food Environment with Diet Quality (GDR) Score

**Model 2: Multivariate Analysis of Features of Home Food Environment with Diet Quality (GDR) Score adjusting for Socio-demographics, Physical Activity and Smoking Status

***Model 3: Multivariate Analysis of Features of Home Food Environment with Diet Quality (GDR) Score adjusting for Socio-demographics, Physical Activity, Smoking Status and aspects of home food environment that include home food availability, accessibility to shops, parental modeling and parenting style

leading to improved diet quality. Conversely, the presence of processed foods at home tends to deteriorate diet quality [5]. However, our study found no significant association between the availability of other healthy and unhealthy food items and diet quality. These contradicting results may be influenced by other factors, such as consumption of food outside home, particularly at school, which can affect diet quality regardless of what is available at home [32].

There was no significant association between parental modeling and parenting style and diet quality. This finding is consistent with a study conducted in adolescent-parent pairs [13] reported that parental modeling and parenting style were not directly associated with adolescent dietary intake, but they may indirectly shape what foods are made available at home. In contrast to previous studies [33], we did not identify any associations of diet quality with the parenting style. Our findings may differ because our study focused on 15–19-year-olds, who are in late adolescence and have greater autonomy in their choices compared to younger adolescents aged 10–14 years [34].

One of the intriguing findings in our study was the negative association between availability of starchy foods, particularly rice, and BMI. Starchy foods, particularly rice, are part of the staple diet in almost every Nepalese household. Everyday availability of certain foods at home is expected to affect the diet pattern and hence the BMI, as seen in a national US study [35]. Starchy food items such as rice contain carbohydrates which are known to contribute to increasing the body weight and would raise the BMI [36] but we found that home availability of a starchy staple in the past day was associated with a lower BMI. This unexpected finding may be attributed to the small number of adolescents (15 out of 678) who reported not having starchy staples available at home, which limits the generalizability of the result. Other dietary factors such as portion sizes and eating frequency that this study did not measure may explain this paradoxical finding. Hence, this finding should be interpreted with caution, as the small sample of respondents without availability of starchy staples at home may not adequately represent broader trends within the population.

Positive parental modeling was associated with a higher BMI. Since the majority of our study participants were either underweight (41%) or had a normal weight (47%), a higher BMI might indicate better health status in our study population. Adolescent Nutrition Survey in Nepal, 2014 indicated that 42.2% of adolescents aged 15–19 years were underweight, 55.9% had a normal weight and only 1.8% were overweight [18] The NDHS 2022 also reported that only 5% of the adolescent girls aged 15–19 years were overweight and less than 1% were obese

**Table 4. Association of Home Food Environment with BMI –Budanilkantha, Nepal, 2022.**

| Features of Home Food Environment | | * Model 1 | | | ** Model 2 | | | *** Model 3 | | |
|---|---|---|---|---|---|---|---|---|---|---|
| Accessibility | n (%) | Coefficient | 95% CI | p-value | Coefficient | 95% CI | p-value | Coefficient | 95% CI | p-value |
| Grocery Shop | | | | | | | | | | |
| 0 minute (Ref) | 67 (9.88) | | | | | | | | | |
| < 10 minutes | 564 (83.19) | −0.46 | (−1.55, 0.62) | 0.40 | −0.52 | (−1.62, 0.57) | 0.35 | −0.99 | (−2.19, 0.22) | 0.11 |
| 10–20 minutes | 39 (5.75) | −0.98 | (−2.68, 0.71) | 0.25 | −1.19 | (−2.91, 0.53) | 0.18 | −1.72 | (−3.62, 0.17) | 0.08 |
| > 20 minutes | 8 (1.18) | −0.82 | (−3.97, 2.32) | 0.61 | −1.13 | (−4.30, 2.05) | 0.49 | −1.83 | (−5.25, 1.59) | 0.29 |
| Fruits Shop | | | | | | | | | | |
| 0 minute (Ref) | 13 (1.92) | | | | | | | | | |
| < 10 minutes | 430 (63.42) | 1.45 | (−0.91, 3.82) | 0.23 | 1.65 | (−0.77, 4.07) | 0.18 | 1.43 | (−1.38, 4.25) | 0.32 |
| 10–20 minutes | 184 (27.14) | 0.74 | (−1.67, 3.15) | 0.55 | 0.85 | (−1.63, 3.32) | 0.50 | 0.39 | (−2.46, 3.26) | 0.78 |
| > 20 minutes | 51 (7.52) | 1.09 | (−1.52, 3.69) | 0.41 | 1.09 | (−1.58, 3.76) | 0.42 | 0.46 | (−2.61, 3.54) | 0.77 |
| Vegetables Shop | | | | | | | | | | |
| 0 minute (Ref) | 24 (3.54) | | | | | | | | | |
| < 10 minutes | 500 (73.75) | 1.61 | (−0.14, 3.37) | 0.07 | 1.82 | (0.04, 3.61) | 0.04 | 1.78 | (−0.26, 3.83) | 0.09 |
| 10–20 minutes | 124 (18.29) | 1.61 | (−0.26, 3.49) | 0.09 | 1.76 | (−0.15, 3.66) | 0.07 | 2.43 | (0.27, 4.59) | 0.03 |
| > 20 minutes | 30 (4.42) | 2.02 | (−0.29, 4.31) | 0.09 | 2.28 | (−0.06, 4.6) | 0.66 | 3.13 | (0.56, 5.72) | 0.02 |
| Street food Vendors | | | | | | | | | | |
| 0 minute (Ref) | 285 (42.04) | | | | | | | | | |
| < 10 minutes | 215 (31.71) | −0.14 | (−0.91, 0.62) | 0.71 | −0.09 | (−0.86, 0.69) | 0.83 | −0.42 | (−1.23, 0.39) | 0.31 |
| 10–20 minutes | 127 (18.73) | −0.33 | (−1.23, 0.57) | 0.47 | −0.35 | (−1.27, 0.57) | 0.45 | −0.51 | (−1.45, 0.43) | 0.29 |
| > 20 minutes | 51 (7.52) | −0.45 | (−1.74, 0.84) | 0.49 | −0.54 | (−1.85, 0.77) | 0.42 | −0.35 | (−1.69, 0.99) | 0.61 |
| Home Food Availability Starchy Staple | | | | | | | | | | |
| No (Ref) | 15 (2.21) | | | | | | | | | |
| Yes | 663 (97.79) | −1.44 | (−4.26, 1.38) | 0.32 | −2.25 | (−5.19, 0.70) | 0.14 | −5.58 | (−10.76, −0.38) | 0.03 |
| Legumes, Nuts & Seeds | | | | | | | | | | |
| No (Ref) | 32 (4.72) | | | | | | | | | |
| Yes | 646 (95.28) | 0.38 | (−1.30, 2.06) | 0.66 | 0.28 | (−1.44, 1.99) | 0.75 | 0.84 | (−2.05, 3.75) | 0.57 |
| Animal-source foods | | | | | | | | | | |
| No (Ref) | 37(5.46) | | | | | | | | | |
| Yes | 641(94.54) | 0.45 | (−1.09, 1.99) | 0.57 | 0.21 | (−1.36, 1.79) | 0.79 | 1.24 | (−0.77, 3.24) | 0.23 |
| Salty or Fried Snacks | | | | | | | | | | |
| No (Ref) | 116 (17.11) | | | | | | | | | |
| Yes | 562 (82.89) | 0.02 | (−0.86, 0.89) | 0.97 | −0.09 | (−1.00, 0.81) | 0.83 | −0.17 | (−1.14, 0.79) | 0.73 |
| Processed meat | | | | | | | | | | |
| No (Ref) | 567 (83.63) | | | | | | | | | |
| Yes | 111 (16.37) | 0.29 | (−0.58, 1.17) | 0.51 | 0.21 | (−0.69, 1.11) | 0.65 | 0.32 | (−0.59, 1.23) | 0.49 |
| Fruits & Vegetables | | | | | | | | | | |
| No (Ref) | 440 (64.90) | | | | | | | | | |
| Yes | 238 (35.10) | −0.83 | (−1.50, −0.15) | 0.02 | −0.65 | (−1.34, 0.05) | 0.07 | −0.66 | (−1.38, 0.06) | 0.07 |
| Fiber containing foods | | | | | | | | | | |
| No (Ref) | 23 (3.39) | | | | | | | | | |
| Yes | 655 (96.61) | 0.14 | (−1.92, 2.21) | 0.89 | 0.02 | (−2.09, 2.14) | 0.99 | 1.82 | (−2.37, 5.99) | 0.39 |
| Sugar sweetened food items | | | | | | | | | | |
| No (Ref) | 30 (4.42) | | | | | | | | | |
| Yes | 648 (95.58) | −0.91 | (−2.65, 0.84) | 0.31 | −0.86 | (−2.63, 0.91) | 0.34 | −0.13 | (−2.43, 2.17) | 0.91 |

*(Continued)*

**Table 4.** (Continued)

| Features of Home Food Environment | | * Model 1 | | | ** Model 2 | | | *** Model 3 | | |
| --- | --- | --- | --- | --- | --- | --- | --- | --- | --- | --- |
| Accessibility | *n* (%) | Coefficient | 95% CI | *p*-value | Coefficient | 95% CI | *p*-value | Coefficient | 95% CI | *p*-value |
| Saturated fat containing foods | | | | | | | | | | |
| No (Ref) | 43 (6.34) | | | | | | | | | |
| Yes | 635 (93.66) | −0.04 | (−1.47, 1.38) | 0.95 | −0.12 | (−1.57, 1.34) | 0.88 | 0.15 | (−1.68, 1.97) | 0.88 |
| Parental Modeling | 678 | 0.15 | (−0.01, 0.31) | 0.06 | 0.18 | (0.01, 0.35) | 0.04 | 0.18 | (0.008, 0.36) | 0.04 |
| Parenting Style | 678 | 0.02 | (−0.04, 0.08) | 0.49 | 0.003 | (−0.06, 0.07) | 0.90 | −0.005 | (−0.07, 0.06) | 0.87 |

Outcome variable: Body Mass Index (BMI)

*Model 1: Bivariate Analysis of Features of Home Food Environment with BMI

**Model 2: Multivariate Analysis of Features of Home Food Environment with BMI adjusting for Socio-demographics, Physical Activity and Smoking Status

***Model 3: Multivariate Analysis of Features of Home Food Environment with BMI adjusting for Socio-demographics, Physical Activity, Smoking Status and aspects of home food environment that include home food availability, accessibility to shops, parental modeling and parenting style

[25]. These data suggest that higher BMI likely indicates better health status in our population, hence, this finding is in expected direction as better eating practices of parents is suggested by positive parental modeling.

The research highlights the critical role of home food environment in shaping diet quality and body mass index among adolescents in urban Nepal. Adolescents face barriers to accessing healthy food options, such as vegetable shops and tend to have lower diet quality suggesting the importance of urban planning initiatives and policy interventions aimed at improving healthy food accessibility. Furthermore, the study emphasizes the significant influence the home environment play. The research contributes valuable insights into the food environment among adolescents within a large urban municipality in Nepal. These findings will aid the researchers, program planners and implementers to provide interventions for devising interventions aimed at improving diet quality among adolescents. Moreover, these findings are helpful. It will also help policy makers to create plans and programs by identifying through evidence the areas where much work needs to be done ultimately developing and implementing food environment policies in a more effective and efficient manner.

There are several strengths to this study. First, to our knowledge, food environment has not been studied before in Nepalese adolescents and this study is first of its kind in Nepal. Second, for characterization of food environment, assessment of home food availability and diet quality, and measurement of parenting style, we have used validated conceptual frameworks of food environment [10] and tools along with expert suggestions, and pretested all tools that were used. Third, though data collection was done through a self-administered questionnaire, in each school, we guided the whole class for each question by writing simplified questions on whiteboard and encouraging them to respond to each question.

This study also has limitations. First, the present study had a cross-sectional design that allowed us to report observations but not causal relationships. Second, there is the risk of respondent bias including social desirability bias when using self-reported measures of dietary intake and food environment, particularly, the actual home food availability might have been lower than the measures obtained. There may have been social desirability biases in the measurement of parental modeling and parenting style as well. Third, while adolescents were asked home food availability questions, it is possible parents would have more accurately responded to these questions since home food inventories are most commonly handled by the parents in household settings. Fourth, our study only included the home food environment, which does not capture the total food environment of adolescents, as they also have their school and social environment among other dimensions of food environment. There are likely many other areas

to intervene in adolescent food environments that were not captured by this study. Fifth, we included those adolescents who were currently enrolled in the school, which limits the generalizability of our findings to those who do not attend school as frequently or are not enrolled.

## Conclusion

This cross-sectional analytical study found that home availability of salty or fried snacks and processed meat negatively affects diet quality, whereas fruits and vegetables positively influence diet quality among adolescents in urban Nepal. Longer walking time from home to vegetable shops and positive parental modeling also positively affect BMI, whereas home availability of starchy staples negatively affects it. Increasing the availability of healthy foods such as fruits and vegetables and limiting unhealthy foods such as processed meat and salty or fried snacks at home may improve the diet quality of adolescents. Also, increasing the easy accessibility of healthy foods may further ensure adolescents' intake. Restricting and monitoring access to unhealthy food items such as sugary drinks, salty or fried snacks among adolescents may help prevent low diet quality. The home food environment and its influence on diet may be unique for adolescents; thus, future research is needed to identify other essential diet influences among this group. This study was limited to the home food environment and hence doesn't capture the wider areas of food environments that adolescents encounter. Given that they are also influenced by factors such as the school food environment, social media, and peer interactions, among others, future studies should consider these dimensions to gain a more comprehensive understanding of their dietary behaviors.

## Supporting information

**S2 Annex: Consent form.**
(WORD)

**S3 Annex: Assent form.**
(WORD)

**S4 Annex: Data Collection form.**
(WORD)

## Acknowledgments

We would like to thank all the participating schools and students of Budanilkantha municipality. We are grateful to Mr. Samip Pandey and Mr. Aditya Raut, who supported us during the data collection and entry. We acknowledge Ms. Anjali Joshi, Mr. Nabin Adhikari, and Mr. Dinesh Timalsena for their support during the data analysis.

## Author contributions

**Conceptualization:** Pragya Sharma, Archana Shrestha.

**Data curation:** Pragya Sharma.

**Formal analysis:** Pragya Sharma, Archana Shrestha.

**Investigation:** Pragya Sharma.

**Methodology:** Pragya Sharma, Archana Shrestha.

**Resources:** Pragya Sharma, Archana Shrestha.

**Software:** Pragya Sharma.

**Supervision:** Archana Shrestha.

**Visualization:** Pragya Sharma, Archana Shrestha.

**Writing – original draft:** Pragya Sharma.

**Writing – review & editing:** Pragya Sharma, Neha Limaye, Rajeeb Kumar Sah, Archana Shrestha.

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
