## [Decision Letter · Decision Letter 0]

28 Dec 2024

PONE-D-24-36721Association of Food Environment with Diet Quality and Body  Mass Index (BMI) of School-going Adolescents in NepalPLOS ONE

Dear Dr. Sharma,

Thank you for submitting your manuscript to PLOS ONE. After careful consideration, we feel that it has merit but does not fully meet PLOS ONE’s publication criteria as it currently stands. Therefore, we invite you to submit a revised version of the manuscript that addresses the points raised during the review process.

We look forward to receiving your revised manuscript.

Kind regards,

Larissa Loures Mendes, Ph.D.

Academic Editor

PLOS ONE

Journal Requirements:

2. Please amend either the title on the online submission form (via Edit Submission) or the title in the manuscript so that they are identical.

3. Please ensure that you refer to Figure 2 in your text as, if accepted, production will need this reference to link the reader to the figure.

4. Please remove all personal information, ensure that the data shared are in accordance with participant consent, and re-upload a fully anonymized data set. 

Reviewers' comments:

Reviewer's Responses to Questions

**Comments to the Author**

1. Is the manuscript technically sound, and do the data support the conclusions?

Reviewer #1: Yes

Reviewer #2: Yes

2. Has the statistical analysis been performed appropriately and rigorously? 

Reviewer #1: Yes

Reviewer #2: Yes

3. Have the authors made all data underlying the findings in their manuscript fully available?

Reviewer #1: Yes

Reviewer #2: Yes

4. Is the manuscript presented in an intelligible fashion and written in standard English?

Reviewer #1: No

Reviewer #2: Yes

5. Review Comments to the Author

Reviewer #1: This study aimed to determine how certain features of the home food environment affect the diet quality and Body Mass Index (BMI) of school-going adolescents. The methods are well described, but some parts need more explanation and discussion. I recommend checking all the text for clarity and consistency of information. Sometimes, the writing makes the text difficult to read. Furthermore, the manuscript is not in the journal's template, which has many details of how to present each section of the manuscript.

Methods

I did not find Table 1, but your work includes many tables and images, more than the journal allows. So, I recommend including Table 1 in the supporting files.

Which software did you use to do the statistical analysis? You need to mention this in your manuscript.

Results

I recommend doing more complete title tables, including the place and year of the study.

I did not find Figure 2. And where is Figure 1? You did not mention it in the text?

In Tables 3 and 4, did you think to characterize your sample by GDR score (categorized) and BMI (categorized)? It would be interesting to see if there is some kind of difference.

Table 4: In the footnote, include all the aspects you included in model 3 of the home food environment. You also need to make clear the outcome variable of your model. You can put the outcome variable in a line above the models.

Based on my comments on Table 4, I recommend revising Table 5. You do not put a footnote in Table 5.

Discussion

I miss citations of studies about the importance of the school environment in adolescents’ food consumption. In the third paragraph, you did not cite one article to discuss this theme; in the next paragraph, when you cited this theme again, you did not mention any previous work.

In your discussion, you cite other works that found similar results. However, I miss you discussing possible reasons for the findings you found, reasons that could justify why you see some patterns in your sample.

You only mention the conceptual framework in the discussion. It should be mentioned in your methods and explain how you use it. Furthermore, citing the supplementary material in the text is recommended so the readers can go to the site and learn that these files exist and learn more about your work.

Reviewer #2: General Considerations

The article has a clear line of reasoning consistent with the components of the text. The procedures used are described in a way that allows for the replication of the study. The instruments appear to be appropriate and utilize reliable methods for data treatment. The conclusions address the objective proposed by the authors. The following corrections are suggested to enhance the value of the work for future publication.

Specific Comments:

Title:

Appropriate and conveys the study’s purpose.

Abstract:

Well-written and consistent. I believe the conclusion could be phrased more effectively.

Introduction:

Clearly presents and connects the facts related to the topic. However, I felt the need for a deeper exploration of the concepts related to the home food environment, as addressed in Gálvez-Espinoza’s (2017) conceptual model, which explores the food environment in Low- and Middle-Income Countries (LMICs). Additionally, it lacks an overview of the nutritional status of adolescents in Nepal, which would better contextualize the findings in the discussion. Moreover, the second paragraph is somewhat confusing. I suggest using this paragraph to discuss the GDR and its relevance to this type of study or why assessing diet quality is important in this population.

Nonetheless, after reading the introduction in its entirety, the motivations for conducting the study are clear.

Methods:

Comprehensive and explanatory, allowing for easy replication of the study. My suggestion is to clarify exactly which sociodemographic variables were used in each regression model, as there are many.

Results:

The results are clearly presented and support the subsequent discussions. Regarding the tables, the captions could be more explicit, in line with the earlier suggestion about detailing the sociodemographic variables used in each regression model.

Additionally, what is the relevance of the data on the "most popular media source"? It is shown in Table 2 but was not used in any analysis. Was there a hypothesis regarding this variable?

Discussion:

The discussion addresses all the results and corroborates the findings with prior literature, but I found the text a bit tedious due to its repetitive structure in each paragraph: presenting the finding in one sentence, bringing in one or two studies that support or contradict it, and then offering hypotheses. I suggest rewriting and better connecting the paragraphs for smoother reading.

I am uncertain about the conclusion that positive parental modeling is associated with higher BMI being a healthy pattern. What is the predominant nutritional status of adolescents in the same age group in the country? Is there any population-based study that supports this conclusion? I did not find references cited in this paragraph.

Conclusion:

Clear and addresses the study question while proposing future studies to address questions this study could not answer.

Regarding the writing, ensure that all parentheses are properly opened and closed.

6. PLOS authors have the option to publish the peer review history of their article (what does this mean? ). If published, this will include your full peer review and any attached files.

**Do you want your identity to be public for this peer review?** For information about this choice, including consent withdrawal, please see our Privacy Policy .

Reviewer #1: No

Reviewer #2: No

---

## [Author Response · Author response to Decision Letter 0]

4 Mar 2025

Response to Comments:

Response: We have checked for templates and inserted line numbers to the document, and we have followed PLOS ONE's style requirements for file naming too.

2. Please amend either the title on the online submission form (via Edit Submission) or the title in the manuscript so that they are identical.

Response: The title of the article has been revised as “Association of food environment with diet quality and Body Mass Index (BMI) of school-going adolescents in Nepal”, in the manuscript as that in online submission form.

3. Please ensure that you refer to Figure 2 in your text as, if accepted, production will need this reference to link the reader to the figure.

Response: The text “S2 Figure 2: Body Mass Index (BMI) Chart of the participants

” is in the Results section (Page number 13).

4. Please remove all personal information, ensure that the data shared are in accordance with participant consent, and re-upload a fully anonymized data set.

Response: All the personal information, including name, contact number and also the participant ID have been removed in the re-uploaded data set.

Response to Reviewers

Reviewer 1:

Methods

Comment: I did not find Table 1, but your work includes many tables and images, more than the journal allows. So, I recommend including Table 1 in the supporting files.

Which software did you use to do the statistical analysis? You need to mention this in your manuscript.

Response: The table is included in supplementary information and named as “S3 Table: Food groups and the corresponding items”.

This way we have renumbered tables and we tables numbered 1-4 in the Result section as compared to 1-5 previously.

We have now included the following sentence under the Statistical Analysis section in the Methods to clarify the software used: "All statistical analyses were conducted using STATA version 14."

Results

Comment: I recommend doing more complete title tables, including the place and year of the study.

Response: We have revised the titles by adding place and year of study as follows:

“Table 1: Socio-demographic characteristics of the participants (n = 678) - Budanilkantha, Nepal, 2022

Table 2: Characteristics of the Participants' Parents (n=678) - Budanilkantha, Nepal, 2022

Table 3: Associations of Home Food Environment with Diet Quality (GDR Score) - Budanilkantha, Nepal, 2022

Table 4: Association of Home Food Environment with BMI - Budanilkantha, Nepal, 2022”

Comment: I did not find Figure 2. And where is Figure 1? You did not mention it in the text?

Response: Figure 1 and Figure 2 are included in supplementary information. Figure 1 is named as S1 Figure 1: Conceptual Framework of the Study and S2 Figure 2: Body Mass Index (BMI) Chart of the participants.

Figure 1 is mentioned in the text in the second last paragraph of Methods section as follows:

“The conceptual model (S1 Figure 1) illustrates the hypothesized relationships among the home food environment, diet quality, BMI and other covariates that include socio-demographic characteristics and lifestyle factors. It also guided the selection of variables for data collection and informed the data analysis by providing a structured approach to examine the interactions among the assessed variables.”

Figure 2 is mentioned in the text as “S2 Figure 2: Body Mass Index (BMI) Chart of the participants” in the Result section.

Comment: In Tables 3 and 4, did you think to characterize your sample by GDR score (categorized) and BMI (categorized)? It would be interesting to see if there is some kind of difference.

Response: The GDR score is a score with a range from 0 to 18 that indicates adherence to global dietary recommendations, which include dietary factors protective against non-communicable diseases. [21] We used GDR score in its continuous variable form during the analyses to capture the level of healthy eating and also avoid arbitrary groupings, as this could introduce potential bias. In addition to this, variables in continuous forms allow greater statistical power in our analyses which is why we kept both of the outcome variables (GDR Score and BMI) in their continuous forms in the analyses.

Comment: Table 3: In the footnote, include all the aspects you included in model 3 of the home food environment.

Response: In table 3, in the footnote, “other aspects of home food environment” has been replaced with “aspects of home food environment that include home food availability, accessibility to shops, parental modeling and parenting style” and the model 3 has been written as follows:

“Model 3: Multivariate Analysis of Features of Home Food Environment with Diet Quality (GDR) Score adjusting for Socio-demographics, Physical Activity, Smoking Status and aspects of home food environment that include home food availability, accessibility to shops, parental modeling and parenting style”

Comment: You also need to make clear the outcome variable of your model. You can put the outcome variable in a line above the models.

Response: In the footnote of table 3, this text has been added: “Outcome variable: Global Dietary Recommendations (GDR) score indicating the diet quality”

In the footnote of table 4, this text has been added: “Outcome variable: Body Mass Index (BMI)”

Comment: Based on my comments on Table 3, I recommend revising Table 4. You do not put a footnote in Table 4.

Response: Footnotes have been added to table 4 as well just like that in table 3 as follows:

“* Model 1: Bivariate Analysis of Features of Home Food Environment with Diet Quality (GDR) Score

** Model 2: Multivariate Analysis of Features of Home Food Environment with Diet Quality (GDR) Score Adjusting for Socio-demographics, Physical Activity and Smoking Status

*** Model 3: Multivariate Analysis of Features of Home Food Environment with Diet Quality (GDR) Score adjusting for Socio-demographics, Physical Activity, Smoking Status and aspects of home food environment that include home food availability, accessibility to shops, parental modeling and parenting style”

Discussion

Comment: I miss citations of studies about the importance of the school environment in adolescents’ food consumption. In the third paragraph, you did not cite one article to discuss this theme; in the next paragraph, when you cited this theme again, you did not mention any previous work.

Response: Thank you for notifying us of this. In the third paragraph of the discussion section, the citation has been added to this text:

“It did not consider the distance from their schools, which could also play a significant role in their eating habits, as adolescents might spend a significant portion of their day at school [29] and could purchase food from vendors near their school instead of near their home as the accessibility of street food in neighborhoods close to schools may have a big impact on their dietary choices. [32]”

In the fourth paragraph of the discussion section, the citation has been added to this text:

“These contradicting results may be influenced by other factors, such as the consumption of food outside the home, particularly at school, which can affect diet quality regardless of what is available at home. [32]”

Comment: In your discussion, you cite other works that found similar results. However, I miss you discussing possible reasons for the findings you found, reasons that could justify why you see some patterns in your sample.

Response: Thank you for the comments. We have addressed it by adding additional texts explaining the reasons for the patterns in our sample. The new texts are pasted below and presented in bold text:

“Adolescents had lower diet quality scores when they lived at a walking distance of less than 10 minutes or greater than 20 minutes to a vegetable shop; this association did not exist for accessibility of fruit shops. Studies [26], [27] have documented that increased accessibility was positively associated with fruit and vegetable intake in children, but this was not seen in our findings. A number of contextual factors may explain this discrepancy. One possible explanation for the lower diet quality associated with vegetable shops farther away is that these places may have fewer overall healthy food alternatives or be located in communities with limited access to fresh produce markets. Furthermore, greater distances to vegetable shops could exacerbate the challenges associated with transportation, time constraints, and economic barriers to purchasing fresh produce. This could lead to a general lack of access to a diverse range of fresh and healthy foods, not only vegetables, resulting in a lower quality diet.

Adolescents also had lower diet quality scores when they lived within 10 minutes or > 20 minutes from street food vendors. However, it's important to note that the study only measured the accessibility to street food shops from the adolescents' homes. It did not consider the distance from their schools, which could also play a significant role in their eating habits, as adolescents might spend a significant portion of their day at school [29] and could purchase food from vendors near their school instead of near their home as the accessibility of street food in neighborhoods close to schools may have a big impact on their dietary choices. [32]

One of the intriguing findings in our study was the negative association between availability of starchy foods, particularly rice, and BMI. Starchy foods, particularly rice, are part of the staple diet in almost every Nepalese household. Everyday availability of certain foods at home is expected to affect the diet pattern and hence the BMI, as seen in a national US study [35]. Starchy food items such as rice contain carbohydrates which are known to contribute to increasing the body weight and would raise the BMI [36] but we found that home availability of a starchy staple in the past day was associated with a lower BMI. This unexpected finding may be attributed to the small number of adolescents (15 out of 678) who reported not having starchy staples available at home, which limits the generalizability of the result. Other dietary factors such as portion sizes and eating frequency that this study did not measure may explain this paradoxical finding. Hence, this finding should be interpreted with caution, as the small sample of respondents without availability of starchy staples at home may not adequately represent broader trends within the population.

Positive parental modeling was associated with a higher BMI. Since the majority of our study participants were either underweight (41%) or had a normal weight (47%), a higher BMI might indicate better health status in our study population. Adolescent Nutrition Survey in Nepal, 2014 indicated that 42.2 % of adolescents aged 15-19 years were underweight, 55.9 % had a normal weight and only 1.8 % were overweight. [18] The NDHS 2022 also reported that only 5% of the adolescent girls aged 15-19 years were overweight and less than 1% were obese. [24] These data suggest that higher BMI likely indicates better health status in our population, hence, this finding is in expected direction as better eating practices of parents is suggested by positive parental modeling.”

Comment: You only mention the conceptual framework in the discussion. It should be mentioned in your methods and explain how you use it. Furthermore, citing the supplementary material in the text is recommended so the readers can go to the site and learn that these files exist and learn more about your work.

Response: Apologies for the confusion. The conceptual framework mentioned in the discussion section is not that of the study but of the food environment. We clarified this further in the second last paragraph of our discussion section and have also provided citation to the supplementary material as follows:

“There are several strengths to this study. First, to our knowledge, the food environment has not been studied before in Nepalese adolescents and this study is first of its kind in Nepal. Second, for our characterization of the food environment, assessment of home food availability and diet quality, and measurement of parenting style, we have used validated conceptual frameworks of the food environment [10] and tools along with expert suggestions, and pretested all tools that were used.” and citation of the conceptual framework has been added too.

Also, details on conceptual framework has been added in the methods section in the second last paragraph as follows:

“A conceptual framework was developed to guide the study, illustrating the relationship between home food environment, diet quality, and BMI among adolescents. The framework allowed identification of the variables and their interaction among each other.

The conceptual model (Figure 1) illustrates the hypothesized relationships among the home food environment, diet quality, BMI and other covariates that include socio-demographic characteristics and lifestyle factors. It also guided the selection of variables for data collection and informed the data analysis by providing a structured approach to examine the interactions among the assessed variables.”

Reviewer 2:

Abstract:

Comment: Well-written and consistent. I believe the conclusion could be phrased more effectively.

Response: Thank you. We have rephrased the conclusion in abstract as follows:

“This study highlights the impact of home food environment on diet quality and BMI among adolescents in urban Nepal. Availability of healthy foods positively affects diet quality, while that of unhealthy items negatively influences it. Future research should explore broader food environmental factors and intervention strategies to improve adolescents’ diet quality.”

Introduction:

Comment: Clearly presents and connects the facts related to the topic. However, I felt the need for a deeper exploration of the concepts related to the home food environment, as addressed in Gálvez-Espinoza’s (2017) conceptual model, which explores the food environment in Low- and Middle-Income Countries (LMICs).

Response: Thank you for the comment. Regarding concepts on food environment, we have accessed info from the paper titled “Concepts and critical perspectives for food environment research: A global framework with implications for action in low- and middle-income countries” and its reference has been provided in our paper. Turner C, Aggarwal A, Walls H, Herforth A, Drewnowski A, Coates J, et al. Concepts and critical perspectives for food environment research: A global framework with implications for action in low- and middle-income countries. Global Food Security. 2018;18: 93–101. doi:10.1016/j.gfs.2018.08.003

We have added the following text in the third paragraph of introduction section to clarify this more:

“The food environment influences food acquisition and consumption of the individuals. It includes external factors such as food availability and price as well as individual determinants like accessibility and affordability.[10]”

Comment: Additionally, it lacks an overview of the nutritional status of adolescents in Nepal, which would better contextualize the findings in the discussion.

Response: Thank you for the comment. We have added the following information on this in the third last paragraph of the introduction section:

Regarding overview of the nutritional status of adolescents in Nepal, this info has been added in the third last paragraph of Introduction section:

“The Nepal Demographic and Health Survey 2022 reports that among adolescent girls aged 15-19 years, 27% are of short stature (indicating stunting), 26% are thin (indicating undernutrition), 5% are overweight, and less than 1% are obese, while among adolescent boys of the same age group, 7% are of short stature (indicating stu

---

## [Editor Report · Decision Letter 1]

10 Mar 2025

Association of food environment with diet quality and Body Mass Index (BMI) of school-going adolescents in Nepal

PONE-D-24-36721R1

Dear Dr. Sharma,

We’re pleased to inform you that your manuscript has been judged scientifically suitable for publication and will be formally accepted for publication once it meets all outstanding technical requirements.

Kind regards,

Larissa Loures Mendes, Ph.D.

Academic Editor

PLOS ONE

Additional Editor Comments (optional):

The revision met all the demands and suggestions of the reviewers and I believe that the article in its current format will contribute to the advancement of the theme of food environments.

---

## [Editor Report · Acceptance letter]

PONE-D-24-36721R1

PLOS ONE

Dear Dr. Sharma,

I'm pleased to inform you that your manuscript has been deemed suitable for publication in PLOS ONE. Congratulations! Your manuscript is now being handed over to our production team.

Kind regards,

on behalf of

Dr. Larissa Loures Mendes

Academic Editor

PLOS ONE